# Effect of chemotherapy on prognosis in patients with primary pancreatic signet ring cell carcinoma: A large real-world study based on machine learning

**Kun Huang**[1]ᴼ*, **Xinzhu Yuan**[2]ᴼ, **Pingwu Zhao**[1], **Yunshen He**[1]

1 Departments of General Surgery, Mian Yang Hospital of Traditional Chinese Medicine, Mianyang, Sichuan, P.R. China, 2 Department of Nephrology, The Second Clinical Medical Institution of North Sichuan Medical College (Nanchong Central Hospital) and Nanchong Key Laboratory of Basic Science & Clinical Research on Chronic Kidney Disease, Nanchong, Sichuan, P.R. China

ᴼ These authors contributed equally to this work.
* mshuangkun@163.com

**Data Availability Statement:** "All relevant data are within the paper and its supporting information files."

## Abstract

### Background

Primary pancreatic signet ring cell carcinoma (PSRCC), an extremely rare histologic variant of pancreatic cancer, has a poor prognosis. This study aimed to investigate the prognostic value of chemotherapy in PSRCC.

### Methods

Patients with PSRCC between 2000 and 2019 were identified using the Surveillance Epidemiology and End Results (SEER) database. The main outcomes in this study were cancer-specific survival (CSS) and overall survival (OS). The baseline characteristics of patients were compared using Pearson's Chi-square test. Kaplan-Meier analysis was used to generate the survival curves. Least absolute shrinkage and selection operator (LASSO), univariate and multivariate Cox regression models, and Random Survival Forest model were used to analyze the prognostic variables for OS and CSS. The variance inflation factors (VIFs) were used to analyze whether there was an overfitting problem.

### Results

A total of 588 patients were identified. Chemotherapy was an independent prognostic factor for OS and CSS, and significantly associated with OS (HR = 0.33, 95% CI = 0.27–0.40, P <0.001) and CSS (HR = 0.32, 95% CI = 0.26–0.39, P <0.001).

### Conclusions

Chemotherapy showed beneficial effects on OS and CSS in patients with PSRCC and should be recommended in clinical practice.

**Funding:** The author(s) received no specific funding for this work.

**Competing interests:** The authors have declared that no competing interests exist.

## Introduction

Pancreatic cancer, the fourth leading cause of cancer death in Europe and the United States [1–4], is a highly malignant tumor with an insidious onset. The incidence and mortality of the cancer showed a continuously increasing trend in China during 1990–2020, and were 8.32/100,000 and 8.48/100,000, respectively in 2020 [5]. On the other hand, the prognosis of pancreatic cancer remains poor with a 5-year survival of 11.5% [6]. Primary pancreatic signet ring cell carcinoma (PSRCC), occurring in less than 1% of pancreatic cancers, is an extremely rare histologic variant with a worse prognosis [7]. Due to the low prevalence, treatment guidelines specific to PSRCC do not exist. In clinical practice, the therapy is guided by existing literature on pancreatic cancer [8]. Surgery and chemotherapy are the main treatment for pancreatic cancer [9]. However, nearly half of the patients had distant metastasis at the time of initial diagnosis, thus losing the opportunity for surgery [10–12]. The value of chemotherapy in PSRCC is still unclear, with the lack of high-quality clinical evidence and large samples of multicenter clinical studies. In this study, we used real-world data from the American Surveillance, Epidemiology, and End Results (SEER) database to investigate the prognostic value of chemotherapy for patients with PSRCC [13].

## Patients and methods

### Patients selection

The SEER-17 Regs Research Plus Data released in April 2022 was retrieved using the SEER*-Stat software version 8.4.0 (https://seer.cancer.gov/seerstat/software/) (National Cancer Institute; National Institutes of Health, USA). Patients were eligible for inclusion, if having the evidence of the primary tumor location stated as 'Pancreas', the code of "ICD-O-3 Hist/behave, malignant" stated as '8490/3: Signet ring cell carcinoma', the years of diagnosis from 2000 through 2019, and PSRCC being the first and only cancer diagnosis; Patients were excluded for missing any information such as radiotherapy records, surgery records, chemotherapy records, survival status, and time. Finally, A total of 588 patients were enrolled in the study. The histologic type was classified using the International Classification of Disease for Oncology, 3rd Edition (ICD-O-3), and the tumor stage categorized using Seer historic stage [14].

### Statistical analysis

The main outcomes in our study were cancer-specific survival (CSS) and overall survival (OS), the specific definition of which referred to studies [13, 15, 16]. Data with continuous variables with nonnormal distributions were presented as M (P25, P75) and those with categorical variables as percentages. The variables were collected for analysis including diagnosis age, race, sex, location of the primary tumor, treatment information, survival time, and survival outcome of patients. The patient baseline characteristics were compared between the chemotherapy group and the non-chemotherapy group using Pearson's chi-square test. Analyses on prognostic factors were performed using univariate and multivariate Cox analyses, least absolute shrinkage and selection operator (LASSO), and Random Survival Forest model, by which the hazard ratios (HRs) with 95% confidence intervals (CIs), nonzero coefficients, and scores of variable importance were calculated, respectively [13, 15]. Survival curves were generated by the Kaplan–Meier method, and differences in survival examined using the log-rank test. The possibility of multicollinearity was estimated using the variance inflation factors (VIFs). If those are less than 10, there was no overfitting problem. The clinical features with a statistical significance in univariate analysis or with clinical significance were selected for further

analyses, but excluded if VIFs of which were greater than 10. Adjusted HRs and 95% CIs were calculated using multivariate Cox proportional hazard models. Subgroup analyses were used to further evaluate the prognostic value of chemotherapy in patients with different clinical features. Stata 16.0/MP and R (version 4.1.1, http://www.r-project.org.) software were used for statistical analysis. A two-sided p-value of less than 0.05 was considered statistically significant.

## Ethics approval and consent to participate

The SEER was public-use data: informed consent was waived, and this is an observational study. The Research Ethics Committee of Mian Yang Hospital of Traditional Chinese Medicine has confirmed that no ethical approval is required.

## Results

### Demographic and clinical characteristics

A total of 588 patients meeting the study criteria were eventually selected, the median age of whom was 67(59, 75) years old. Of the 588 patients, 259 (44.05%) were female, and 329 (55.95%) male; 267 (45.41%) underwent chemotherapy and 321(54.59%) did not. Patients' demographics and tumor characteristics stratified by chemotherapy are summarized in Table 1. Using Pearson $\chi 2$ -test, significant differences between the two groups were observed concerning age ($p < 0.001$), tumor size (p = 0.021), marital status ($p = 0.001$), liver metastasis ($p = 0.002$), and radiotherapy ($p < 0.001$) (Table 1).

### Survival outcome analysis

The median follow-up was 3 (1,9) months. Of the 267patients in the chemotherapy group, 246 died (92.1%), including 233 tumor-related deaths (87.3%); Of the 321 patients in the non-chemotherapy group, 310 died (96.6%), including 294 tumor-related deaths (91.6%). The estimated 1-year OS of the chemotherapy group and the non-chemotherapy group were 31.2% and 13.4%, respectively and the estimated 1-year CSS were 32.4% and 14.0%, respectively. There were significant differences in OS and CSS between the two groups (all $P < 0.05$), suggesting chemotherapy could increase the OS and CSS in patients with PSRCC (Fig 1).

**Univariate and multivariate Cox regression model.** To identify more independent prognostic factors for OS and CSS, the univariate and the multivariate Cox regression analysis were used. In the univariate Cox regression model, we found that age, tumor location, tumor size, marital status, tumor stage, grade, liver metastasis, radiotherapy, surgery, and chemotherapy were associated with the OS and the CSS (all $P < 0.05$) (Table 2).

Further, the aforementioned variables were analyzed in a multivariate Cox regression model, to identify the independent prognostic factors for OS and CSS. The analysis shows that age, tumor location, marital status, tumor stage, grade, liver metastasis, surgery, and chemotherapy were associated with the OS and the CSS (all $P < 0.05$). Meanwhile, chemotherapy significantly improved the OS (HR = 0.34, 95% CI = 0.28 to 0.41, P <0.001) and CSS (HR = 0.33, 95% CI = 0.27 to 0.41, P <0.001), after adjusting for age, tumor location, marital status, tumor stage, grade, liver metastasis, and surgery. The details are shown in Fig 2.

**LASSO regression model.** LASSO regression known to be able to remove unimportant variables via the regression coefficients penalizing the size of the parameters has been extended and broadly applied to the Cox proportional hazard regression model for survival analysis [17]. The coefficient estimates can be shrunk toward zero, with the degree of shrinkage dependent on an additional parameter, λ (Fig 3A and 3C). To determine the optimal values for λ, 10-fold cross-validation was used, and we chose λ via the1-SE criteria [18]. Finally, a λ value

**Table 1. Baseline characteristics of patients with PSRCC.**

| Characteristic | Total (n = 588) | Chemotherapy | | P |
| --- | --- | --- | --- | --- |
| | | **No (n = 321)** | **Yes (n = 267)** | |
| **Age, n (%), y** | | | | < 0.001 |
| <60 | 155 (26.4) | 62 (19.3) | 93 (34.8) | |
| ≥60 | 433 (73.6) | 259 (80.7) | 174 (65.2) | |
| **Sex, n (%)** | | | | 0.064 |
| Female | 259 (44.0) | 153 (47.7) | 106 (39.7) | |
| Male | 329 (56.0) | 168 (52.3) | 161 (60.3) | |
| **Race, n (%)** | | | | 0.278 |
| White | 486 (82.7) | 258 (80.4) | 228 (85.4) | |
| Black | 60 (10.2) | 37 (11.5) | 23 (8.6) | |
| Other | 42 (7.1) | 26 (8.1) | 16 (6.0) | |
| **Tumor size, n (%), cm** | | | | 0.021 |
| ≤4cm | 148 (25.2) | 69 (21.5) | 79 (29.6) | |
| >4cm | 339 (57.7) | 187 (58.3) | 152 (56.9) | |
| Unknown | 101 (17.2) | 65 (20.2) | 36 (13.5) | |
| **Tumor location, n (%)** | | | | 0.178 |
| Head | 282 (48.0) | 148 (46.1) | 134 (50.2) | |
| Body/tail | 143 (24.3) | 74 (23.1) | 69 (25.8) | |
| Unknown | 163 (27.7) | 99 (30.8) | 64 (24.0) | |
| **Marital status, n (%)** | | | | 0.001 |
| Married | 339 (57.7) | 163 (50.8) | 176 (65.9) | |
| Unmarried/divorce | 225 (38.3) | 142 (44.2) | 83 (31.1) | |
| Unknown | 24 (4.1) | 16 (5.0) | 8 (3.0) | |
| **Tumor stage, n (%)** | | | | 0.603 |
| Localized/regional | 147 (25.0) | 75 (23.4) | 72 (27.0) | |
| Distant | 333 (56.6) | 186 (57.9) | 147 (55.1) | |
| Unknown | 108 (18.4) | 60 (18.7) | 48 (18.0) | |
| **Liver metastasis, n (%)** | | | | 0.002 |
| No | 149 (25.3) | 65 (20.2) | 84 (31.5) | |
| Yes | 117 (19.9) | 60 (18.7) | 57 (21.3) | |
| Unknown | 322 (54.8) | 196 (61.1) | 126 (47.2) | |
| **Grade, n (%)** | | | | 0.486 |
| I/II | 33 (5.6) | 15 (4.7) | 18 (6.7) | |
| III/IV | 234 (39.8) | 132 (41.1) | 102 (38.2) | |
| Unknown | 321 (54.6) | 174 (54.2) | 147 (55.1) | |
| **Surgery, n (%)** | | | | 0.112 |
| No | 482 (82.0) | 271 (84.4) | 211 (79.0) | |
| Yes | 106 (18.0) | 50 (15.6) | 56 (21.0) | |
| **Radiotherapy, n (%)** | | | | < 0.001 |
| No | 553 (94.0) | 317 (98.8) | 236 (88.4) | |
| Yes | 35 (6.0) | 4 (1.2) | 31 (11.6) | |

(for OS) of 0.2, with log ($\lambda$), -1.6, and a $\lambda$ value (for CSS) of 0.19, with log ($\lambda$), -1.6 was chosen (Fig 3B and 3D). The variables with nonzero coefficients including chemotherapy ($\beta$ = -0.24118), surgery ($\beta$ = -0.52625), and tumor stage ($\beta$ = 0.00027) were independent influencing factors for OS and chemotherapy ($\beta$ = -0.248), surgery ($\beta$ = -0.537) and tumor stage ($\beta$ = 0.013) for CSS, showed by the results of the LASSO regression model.

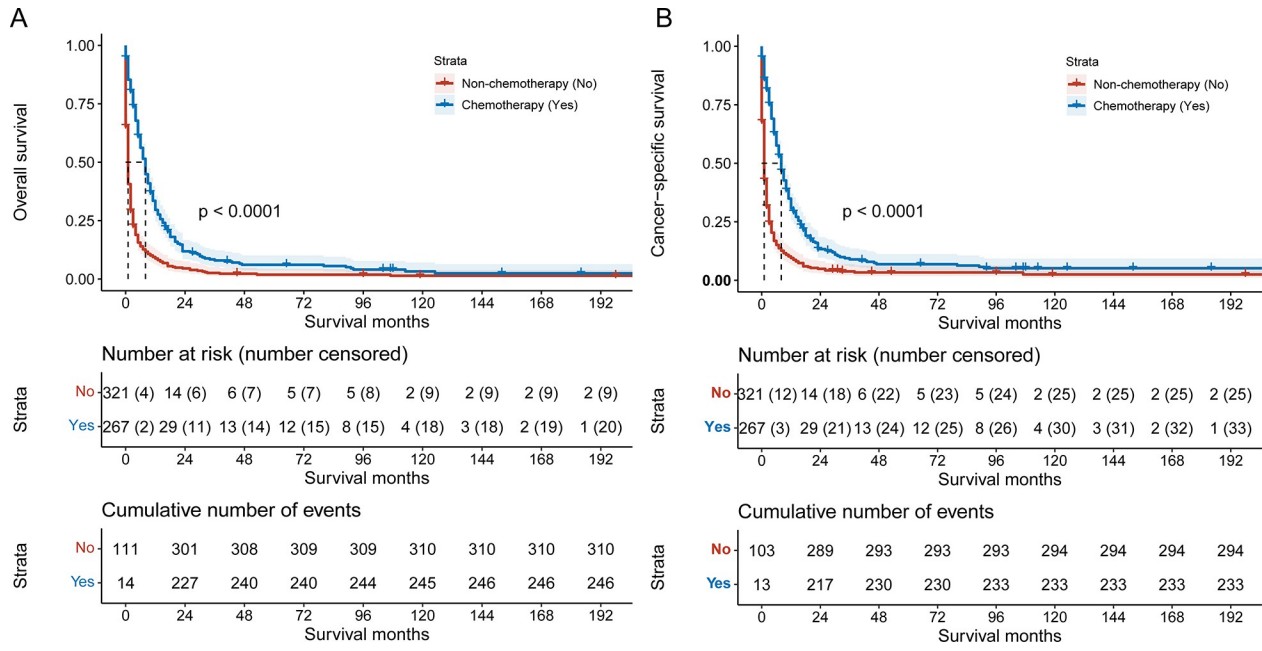

**Fig 1.** OS (A) and CSS (B) curves of PSRCC patients.

**Random Survival Forest model.** Random Survival Forest model, a machine learning algorithm with high robustness and without the restriction of Proportional Hazards Assumption, can prevent over-fitting problems via two random sampling processes. Here, we calculated scores of variable importance for OS and CSS, using the Random Survival Forest model. The results also indicated chemotherapy was an independent influencing factor for OS and CSS, with an importance score of 0.053 and 0.055, respectively (Fig 4).

**Survival analysis for subgroups.** Subgroup analyses using the Cox model were conducted to further determine the effect of chemotherapy for OS and CSS in patients with different features. we found that chemotherapy increased the OS and CSS in all subgroups. The results of the analyses are shown in Fig 5.

## Discussion

As a subtype of pancreatic ductal adenocarcinoma, PSRCC is very rare [8]. To our best knowledge, only 21 cases were reported in the literature [8, 19–35]. The value of chemotherapy was still unclear for the lack of high-quality clinical evidence and large samples of multicenter clinical studies. For this purpose, we used real-world data from the SEER database to investigate the value of chemotherapy for PSRCC patients. Previous studies have confirmed that chemotherapy has well-established survival benefits for patients with pancreatic cancer [36–39]. Yet, the value of chemotherapy for PSRCC remains unclear.

In the present study based on the largest sample, we showed for the first time in PSRCC that chemotherapy was significantly associated with OS (HR = 0.33, 95% CI = 0.27–0.4, P <0.001) and CSS (HR = 0.32, 95% CI = 0.26–0.39, P <0.001), that an independent factor influencing prognosis, and that could independently improve the OS and the CSS.

Radojkovic et al. [24] presented a PSRCC patient with a good response to neoadjuvant chemotherapy (after a 3-month neoadjuvant chemotherapy course with gemcitabine alone, the tumor in the head of the pancreas with 4.5 cm in the largest diameter regressed to 1.5 cm in largest diameter). Our findings, to some extent, were further confirmed by Radojkovic et al.

**Table 2. Univariate Cox regression analysis of CSS and OS in PSRCC patients.**

| Variables | OS | | CSS | |
|---|---|---|---|---|
| | HR (95%CI) | *P* | HR (95%CI) | *P* |
| **Age, y** | | | | |
| <60 | Reference | | Reference | |
| ≥60 | 1.32 [1.09, 1.59] | 0.005 | 1.27 [1.05, 1.55] | 0.015 |
| **Sex** | | | | |
| Female | Reference | | Reference | |
| Male | 1.02 [0.86, 1.21] | 0.825 | 1.04 [0.87, 1.24] | 0.661 |
| **Race** | | | | |
| White | Reference | | Reference | |
| Black | 1.06 [0.81, 1.40] | 0.673 | 1.06 [0.80, 1.41] | 0.665 |
| Other | 1.02 [0.73, 1.42] | 0.893 | 1.03 [0.73, 1.44] | 0.879 |
| **Tumor location** | | | | |
| Head | Reference | | Reference | |
| Body/tail | 1.29 [1.05, 1.58] | 0.017 | 1.28 [1.04, 1.58] | 0.022 |
| Unknown | 1.80 [1.47, 2.20] | <0.001 | | |
| **Tumor size, cm** | | | | |
| ≤4cm | Reference | | Reference | |
| >4cm | 1.33 [1.09, 1.62] | 0.005 | 1.34 [1.09, 1.65] | 0.005 |
| Unknown | 1.94 [1.49, 2.51] | <0.001 | 1.93 [1.48, 2.52] | <0.001 |
| **Marital status** | | | | |
| Married | Reference | | Reference | |
| Unmarried/divorce | 1.28 [1.08, 1.53] | 0.005 | 1.30 [1.08, 1.55] | 0.004 |
| Unknown | 1.30 [0.85, 1.98] | 0.227 | 1.24 [0.80, 1.93] | 0.341 |
| **Tumor stage** | | | | |
| Localized/regional | Reference | | Reference | |
| Distant | 2.28 [1.85, 2.80] | <0.001 | 2.31 [1.86, 2.85] | <0.001 |
| Unknown | 1.63 [1.24, 2.13] | <0.001 | 1.61 [1.22, 2.13] | 0.001 |
| **Grade** | | | | |
| I/II | Reference | | Reference | |
| III/IV | 1.86 [1.25, 2.76] | 0.002 | 2.11 [1.37, 3.25] | 0.001 |
| Unknown | 2.67 [1.80, 3.97] | <0.001 | 2.99 [1.94, 4.60] | <0.001 |
| **Liver metastasis** | | | | |
| No | Reference | | Reference | |
| Yes | 1.78 [1.38, 2.30] | <0.001 | 1.90 [1.46, 2.47] | <0.001 |
| Unknown | 1.56 [1.27, 1.92] | <0.001 | 1.63 [1.32, 2.02] | <0.001 |
| **Radiotherapy** | | | | |
| No | Reference | | Reference | |
| Yes | 0.34 [0.23, 0.49] | <0.001 | 0.33 [0.22, 0.48] | <0.001 |
| **Surgery** | | | | |
| No | Reference | | Reference | |
| Yes | 0.31 [0.24, 0.39] | <0.001 | 0.30 [0.23, 0.39] | <0.001 |
| **Chemotherapy** | | | | |
| No | Reference | | Reference | |
| Yes | 0.41 [0.34, 0.49] | <0.001 | 0.41 [0.34, 0.49] | <0.001 |

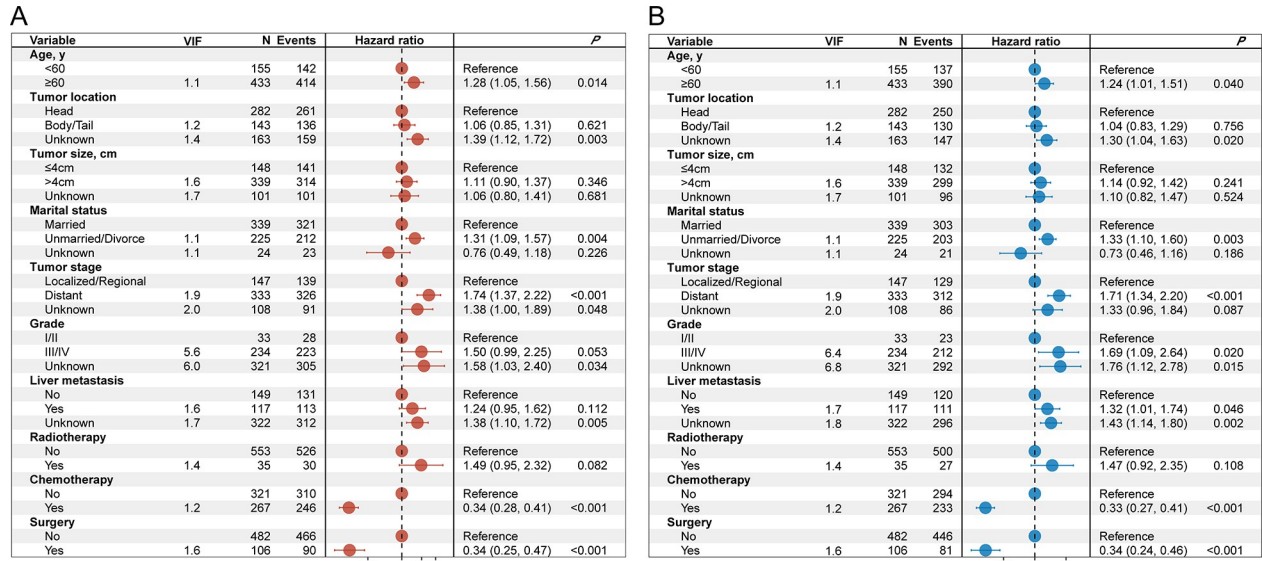

**Fig 2.** The forest plots for multivariate Cox regression analysis of OS (A) and CSS (B) in patients with PSRCC. VIF = variance inflation factor.

[24]. Similar to our study, a recent study indicated that chemotherapy enhanced the CSS (HR = 0.549, 95% CI = 0.413–0.728, P <0.001) for bladder signet ring cell carcinoma [14], further corroborating our findings. In addition, Hugen et al. [40] also found that adjuvant chemotherapy is associated with improving survival in colorectal signet-ring cell carcinoma patients. Furthermore, in the study by Cai et al. [41], it was also found that chemotherapy was significantly associated with OS (HR = 0.54, 95% CI = 0.45–0.65, P <0.0001). The studies by Hugen et al. [40] and Cai et al. [41] were consistent with our findings.

Yet, in contrast to the findings described above, gastric signet ring cell carcinoma and colorectal signet ring cell carcinoma were not sensitive to chemotherapy in some studies [42–45], which may be explained by the different chemotherapy protocols and the unique biological behavior. For gastric signet ring cell carcinoma, previous chemotherapy protocol was largely based on treatment with 5-fluorouracil and platinum-based therapies, but the subsequent study suggests that gastric signet ring cell carcinoma may be uniquely chemosensitive to taxane-based therapy [46]. Furthermore, Thymidylate Synthase, the key enzyme for DNA synthesis pathways, is inhibited by 5-Fluorouracil. A study by Cabibi et al. [43] showed that colorectal signet ring cell carcinoma was negative for thymidylate synthase, and it may be one of the reasons for the insensitivity to chemotherapy.

The following limitations of this study should be considered. First, there was selection bias in this study, for being a retrospective analysis [14]. Second, due to the limitations of the SEER database, the concrete chemotherapy regimen and the details of local regional recurrence data were not available, to impact the survival. Third, as an important factor for clinical decision-making and survival, the performance status of each patient was not provided in the SEER database. Our study has several strengths. First, we independently used univariate and multivariate Cox regression, LASSO, and Random Survival Forest model to analyze the prognostic factors for survival and also used subgroup analysis to adjust all other variables affecting the prognosis, making our conclusions more reliable and stable. Second, in comparison to a single institution, the SEER database has access to a much larger cohort of patients. To the best of

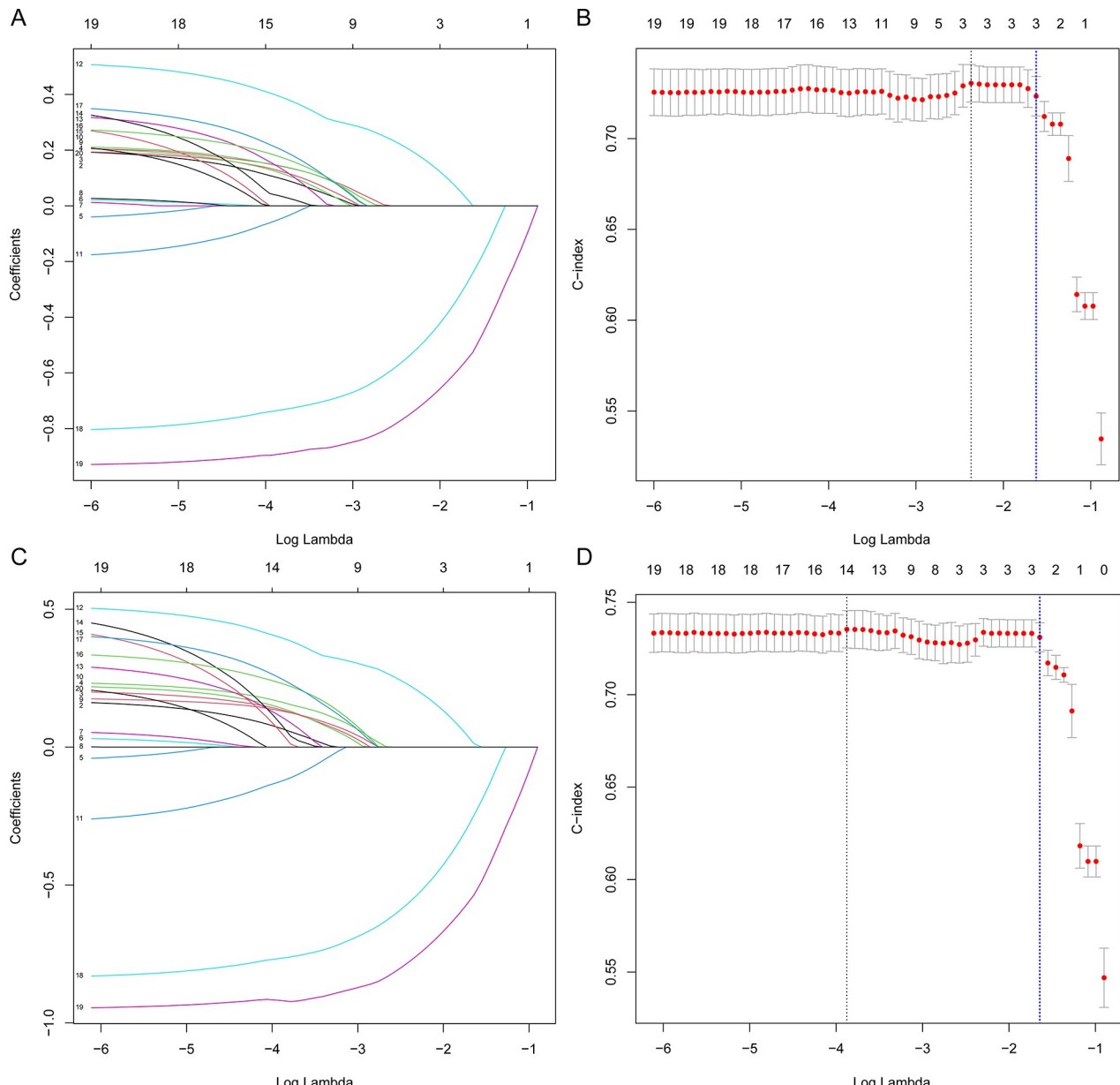

**Fig 3.** Feature selection based on LASSO regression (A and B for OS; C and D for CSS). LASSO coefficient profiles of the candidate variables, and a coefficient profile plot was produced against the log (λ) sequence; With the increment of the log (λ), the coefficient estimates shrink toward zero (A, C). tuning parameter (λ) selection in the LASSO model used 10-fold cross-validation via the 1-SE criteria, and the C-index curve was plotted versus log(λ); a λ value (for OS) of 0.2, with log (λ), -1.6 and a λ value (for CSS) of 0.19, with log (λ), -1.6 was chosen (blue dash line in B, D).

our knowledge, this study included the largest sample for evaluating the value of chemotherapy in patients with PSRCC to date. Third, our study included 588 patients with 566 deaths for OS and 527 deaths for CSS. The large sample size and number of deaths provided sufficient power for the analyses.

In conclusion, Patients with PSRCC can benefit from chemotherapy, to be recommended for patients with PSRCC.

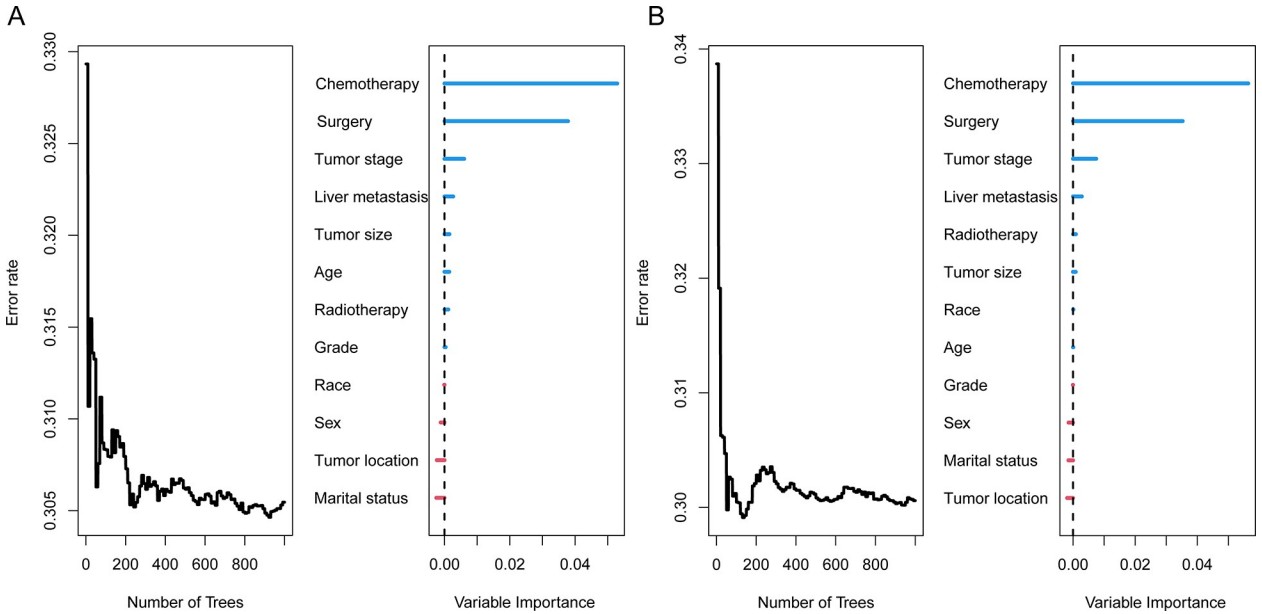

**Fig 4.** Scores of variable importance for OS (A) and CSS (B) using Random Survival Forest model.

**Fig 5.** The forest plots of the influence of chemotherapy on survival in different subgroups (A for OS and B for CSS).

## Supporting information

**S1 Table. The minimal data.**
(XLSX)

## Acknowledgments

We thank the open access to the database from SEER.

## Author Contributions

**Conceptualization:** Kun Huang.

**Data curation:** Xinzhu Yuan.

**Formal analysis:** Xinzhu Yuan.

**Methodology:** Kun Huang, Xinzhu Yuan.

**Visualization:** Yunshen He.

**Writing – original draft:** Kun Huang, Yunshen He.

**Writing – review & editing:** Pingwu Zhao.

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
