## [Decision Letter · Decision Letter 0]

5 Mar 2024

PONE-D-24-04991Effect of chemotherapy on prognosis in patients with primary pancreatic signet ring cell carcinoma: A large real-world study based on machine learningPLOS ONE

Dear Dr. Huang,

Thank you for submitting your manuscript to PLOS ONE. After careful consideration, we feel that it has merit but does not fully meet PLOS ONE’s publication criteria as it currently stands. Therefore, we invite you to submit a revised version of the manuscript that addresses the points raised during the review process.

We look forward to receiving your revised manuscript.

Kind regards,

Filomena de Nigris, Ph.D.

Academic Editor

PLOS ONE

Journal Requirements:

2. In this instance it seems there may be acceptable restrictions in place that prevent the public sharing of your minimal data. However, in line with our goal of ensuring long-term data availability to all interested researchers, PLOS’ Data Policy states that authors cannot be the sole named individuals responsible for ensuring data access (http://journals.plos.org/plosone/s/data-availability#loc-acceptable-data-sharing-methods).

Reviewers' comments:

Reviewer's Responses to Questions

**Comments to the Author**

1. Is the manuscript technically sound, and do the data support the conclusions?

Reviewer #1: Yes

Reviewer #2: Partly

2. Has the statistical analysis been performed appropriately and rigorously? 

Reviewer #1: Yes

Reviewer #2: I Don't Know

3. Have the authors made all data underlying the findings in their manuscript fully available?

Reviewer #1: Yes

Reviewer #2: Yes

4. Is the manuscript presented in an intelligible fashion and written in standard English?

Reviewer #1: Yes

Reviewer #2: Yes

5. Review Comments to the Author

Reviewer #1: The manuscript is well presented, the topics well described. The tables and graphs are clear and sufficient. Only a few aspects would need to be revised by the authors:

• The work has no innovative features. Authors are requested to emphasize the new aspects considered in comparison with the following work: Nie, D., Lan, Q., Huang, Y. et al. Epidemiology and prognostic analysis of patients with pancreatic signet ring cell carcinoma: a population-based study. BMC Gastroenterol 22, 458 (2022). 10.1186/s12876-022-02543-z

• Authors should specify how they intend to overcome the bias of their study in the discussion.

• Authors should review the bibliography (ex: 4-5)

Reviewer #2: In the paper by Yunshen He et al., the role of chemotherapy as a prognostic factor of primary pancreatic singet ring cell carcinoma, a rare pancreatic ductal adenocarcinoma, is investigated through a statistical analysis. the paper is well written but:

- The introduction is too short and does not delve into any topic about pancretic cancer or PSRCC. It is not satisfactory for a student reader and very poor for an academic reader.

- Statistical analysis does not make any histological classification - Statistical analysis does not mention the different treatment groups, but in general chemotherapy it is not easy to understand if there is an effect of radiotherapy. The author should be able to take the effect of radiotherapy or a particular radiotherapy protocol in combination or individually.

Although the idea of the proposed analysis may be interesting, it is presented in a very superficial way, leaving unresolved many aspects about the variability of treatments, staging, etc.

6. PLOS authors have the option to publish the peer review history of their article (what does this mean?). If published, this will include your full peer review and any attached files.

Reviewer #1: No

Reviewer #2: No

---

## [Author Response · Author response to Decision Letter 0]

22 Mar 2024

23th Mar, 2024

PONE-D-24-04991

Effect of chemotherapy on prognosis in patients with primary pancreatic signet ring cell carcinoma: A large real-world study based on machine learning.

Dear editors,

 Thank you for your thoughtful review of our manuscript and the helpful comments from the reviewers. We agree with the reviewers’ suggestions. Here below a point-by-point reply to the comments. Major changes are marked in the revised manuscript. Hopefully you will find the revised manuscript improved and suitable for publication in PLOS ONE.

Reviewer 1

Q1:

The manuscript is well presented, the topics well described. The tables and graphs are clear and sufficient. Only a few aspects would need to be revised by the authors: The work has no innovative features. Authors are requested to emphasize the new aspects considered in comparison with the following work: Nie, D., Lan, Q., Huang, Y. et al. Epidemiology and prognostic analysis of patients with pancreatic signet ring cell carcinoma: a population-based study. BMC Gastroenterol 22, 458 (2022). 10.1186/s12876-022-02543-z; Authors should specify how they intend to overcome the bias of their study in the discussion.

Response:

We have meticulously reviewed the study by Nie et al., which centers on investigating the epidemiology of PSRCC. However, the content regarding prognostic analysis is relatively limited, and it does not specifically target the analysis of the value of chemotherapy. Furthermore, the outcome measures in this article are limited to Overall Survival (OS) only. In addition, the baseline characteristics of patients are not comprehensively presented in the comparison between chemotherapy and non-chemotherapy groups. In contrast, in this study, we aimed to specially investigate the prognostic value of chemotherapy in PSRCC. The patients’ demographics and tumor characteristics stratified by chemotherapy are detailedly summarized in Table 1. Also, the main outcomes in this study were cancer-specific survival (CSS) and OS, which enhances the reliability of the results. Further, we independently used univariate and multivariate Cox regression, LASSO, and Random Survival Forest model to analyze the prognostic factors for survival and also used subgroup analysis to adjust all other variables affecting the prognosis, making our conclusions more reliable and stable.

Notably, in addition to the aforementioned aspects, we have also utilized machine learning algorithms to optimize the methodology. 

Q2:

Authors should review the bibliography (ex: 4-5).

Response:

We have meticulously reviewed all the references in the full manuscript and confirmed their accuracy. For references 4 and 5, we have downloaded the PDF documents and uploaded them as supplementary materials.

Reviewer 2

Q1:

The introduction is too short and does not delve into any topic about pancretic cancer or PSRCC. It is not satisfactory for a student reader and very poor for an academic reader.

Response:

We have made relevant additions and revisions to the introduction section.

Q2:

Statistical analysis does not make any histological classification 

Response:

PSRCC is an extremely rare histologic variant of pancreatic cancers with the code of “ICD‑O‑3 Hist/behave, malignant” stated as ‘8490/3: Signet ring cell carcinoma’. In this study, we exclusively focus on a specific histological type of pancreatic cancers to investigate the influence of chemotherapy on prognosis. Consequently, the statistical analysis does not involve any other histological classification.

Q3:

Statistical analysis does not mention the different treatment groups, but in general chemotherapy it is not easy to understand if there is an effect of radiotherapy. The author should be able to take the effect of radiotherapy or a particular radiotherapy protocol in combination or individually. Although the idea of the proposed analysis may be interesting, it is presented in a very superficial way, leaving unresolved many aspects about the variability of treatments, staging, etc.

Response:

In our study, we not only utilized the standard univariate and multivariate Cox regression models and LASSO regression model for prognostic variable analysis but also employed the Random Survival Forest model and Survival Analysis for Subgroups. The Random Survival Forest model is advantageous as it lacks the constraint of the Proportional Hazards Assumption and prevents overfitting through its dual random sampling processes. On the other hand, Survival Analysis for Subgroups allows for the observation of chemotherapy's impact on patient prognosis at various subgroup levels after adjusting for the influence of other variables. This comprehensive statistical approach enhances the robustness of our findings and provides a nuanced understanding of how chemotherapy may affect outcomes within specific patient populations. ( Fig 5)

---

## [Decision Letter · Decision Letter 1]

9 Apr 2024

Effect of chemotherapy on prognosis in patients with primary pancreatic signet ring cell carcinoma: A large real-world study based on machine learning

PONE-D-24-04991R1

Dear Dr Huang

We’re pleased to inform you that your manuscript has been judged scientifically suitable for publication and will be formally accepted for publication once it meets all outstanding technical requirements.

Kind regards,

Filomena de Nigris, Ph.D.

Academic Editor

PLOS ONE

Additional Editor Comments (optional):

Reviewers' comments:

Reviewer's Responses to Questions

**Comments to the Author**

1. If the authors have adequately addressed your comments raised in a previous round of review and you feel that this manuscript is now acceptable for publication, you may indicate that here to bypass the “Comments to the Author” section, enter your conflict of interest statement in the “Confidential to Editor” section, and submit your "Accept" recommendation.

Reviewer #1: All comments have been addressed

Reviewer #2: All comments have been addressed

2. Is the manuscript technically sound, and do the data support the conclusions?

Reviewer #1: Yes

Reviewer #2: Yes

3. Has the statistical analysis been performed appropriately and rigorously? 

Reviewer #1: Yes

Reviewer #2: I Don't Know

4. Have the authors made all data underlying the findings in their manuscript fully available?

Reviewer #1: Yes

Reviewer #2: Yes

5. Is the manuscript presented in an intelligible fashion and written in standard English?

Reviewer #1: Yes

Reviewer #2: Yes

6. Review Comments to the Author

Reviewer #1: The authors have responded clearly and comprehensively to the comments raised in the first revision, therefore the manuscript is suitable for publication

Reviewer #2: the author's answers are exhaustive within the scope of the paper's objective even if some remain perplexed about the validity of the paper's idea.

7. PLOS authors have the option to publish the peer review history of their article (what does this mean?). If published, this will include your full peer review and any attached files.

Reviewer #1: No

Reviewer #2: No

---

## [Editor Report · Acceptance letter]

30 Apr 2024

PONE-D-24-04991R1 

PLOS ONE

Dear Dr. Huang, 

I'm pleased to inform you that your manuscript has been deemed suitable for publication in PLOS ONE. Congratulations! Your manuscript is now being handed over to our production team.

Kind regards, 

on behalf of

Prof. Filomena de Nigris 

Academic Editor

PLOS ONE